# Combined Effects of Fibroblast Growth Factor-2 and Carbonate Apatite Granules on Periodontal Healing: An In Vivo and In Vitro Study

**DOI:** 10.3390/biomedicines12081664

**Published:** 2024-07-25

**Authors:** Naoki Miyata, Shinta Mori, Tasuku Murakami, Takahiro Bizenjima, Fumi Seshima, Kentaro Imamura, Atsushi Saito

**Affiliations:** 1Department of Periodontology, Tokyo Dental College, Chiyoda-ku, Tokyo 1010061, Japan; miyatanaoki@tdc.ac.jp (N.M.); morishinta@tdc.ac.jp (S.M.); murakamitasuku@tdc.ac.jp (T.M.); seshimaf@tdc.ac.jp (F.S.); imamurakentarou@tdc.ac.jp (K.I.); 2Oral Health Science Center, Tokyo Dental College, Chiyoda-ku, Tokyo 1010061, Japan; 3Chiba Dental Center, Tokyo Dental College, Mihama-ku, Chiba 2618502, Japan; bizenjimatakahiro@tdc.ac.jp

**Keywords:** fibroblast growth factor-2, carbonate apatite granules, periodontal regeneration, bone graft

## Abstract

The aim of this study was to investigate in vivo and in vitro the effectiveness of the use of fibroblast growth factor (FGF)-2 with carbonate apatite (CO_3_Ap) on periodontal healing. Periodontal defects created in the maxillary first molars in rats were treated with FGF-2, CO_3_Ap, FGF-2 + CO_3_Ap or left unfilled. Healing was evaluated using microcomputed tomography, histological, and immunohistochemical analyses. In vitro experiments were performed to assess cellular behaviors and the expression of osteoblastic differentiation markers in MC3T3-E1 cells. At 4 weeks, the bone volume fraction in the FGF-2 + CO_3_Ap group was significantly greater than that in the CO_3_Ap group, but there was no significant difference from the FGF-2 group. The FGF-2 + CO_3_Ap group demonstrated greater new bone compared with the FGF-2 or CO_3_Ap group. The FGF-2 + CO_3_Ap group showed greater levels of osteocalcin-positive cells compared with the CO_3_Ap group, but there was no significant difference from the FGF-2 group. In vitro, the FGF-2 + CO_3_Ap group exhibited a greater extent of cell attachment and more elongated cells compared with the CO_3_Ap group. Compared with the CO_3_Ap group, the FGF-2 + CO_3_Ap group showed significantly higher viability/proliferation, but the expressions of *Runx2* and *Sp7* were reduced. The results indicated that the use of FGF-2 with CO_3_Ap enhanced healing in the periodontal defects. FGF-2 promoted cell attachment to and proliferation on CO_3_Ap and regulated osteoblastic differentiation, thereby contributing to novel bone formation.

## 1. Introduction

Periodontitis is an inflammatory disease that affects susceptible individuals, in part due to reciprocally reinforced interactions between the pathogenic microbiome and host immune and inflammatory responses [1]. Periodontitis can lead to progressive destruction of the periodontal tissues (gingiva, periodontal ligament, cementum, and alveolar bone) and eventual loss of teeth, thus compromising eating ability and esthetics and affecting life quality [1,2]. In cases of moderate to severe periodontitis, surgical intervention is often needed in addition to the non-surgical therapy. The goal of periodontal therapy is to regenerate the lost periodontal structures, and various methods of periodontal regenerative therapy are used for this purpose. Three major factors are important for periodontal tissue engineering: signaling molecules, scaffolds, and stem cells [3,4]. Signaling molecules promote cell proliferation and differentiation, and scaffolds provide three-dimensional structures for host cells to support tissue regeneration [5].

Fibroblast growth factor-2 (FGF-2) is a biological agent that promotes the migration and proliferation of periodontal ligament-derived cells (PDLCs), which play an important role in periodontal regeneration [3]. In Japan, a commercial formulation of 0.3% recombinant human FGF-2 (rhFGF-2) was formally approved as a drug for periodontal regeneration, following extensive clinical trials [6,7,8]. The rhFGF-2 formulation has been clinically used for periodontal regenerative therapy [9,10,11,12,13,14].

Bone graft materials have been used as a scaffold for periodontal regenerative therapy. Currently available bone graft materials include autologous, allogenic, xenogeneic, and artificial bone materials. Ideally, bone graft materials should have osteoconductive, osteoinductive, and osteogenic properties. Autologous bone is reported to have all three properties. However, the amount of bone that can be collected is limited, and harvesting involves patient morbidity. Allogenic bone has a concern for the risk of infection from unknown sources. Xenogeneic bone such as demineralized bovine bone mineral (DBBM) has been shown to have osteoconductive properties [15]. DBBM is absorbed slowly and thus functions long term as a scaffold [16]. There are various artificial bone substitutes such as beta-tricalcium phosphate (*β*-TCP) and hydroxyapatite (HA). Artificial bone is safer than other materials in terms of risk of infection and has no restrictions on quantity. Therefore, it may be suitable for periodontal regenerative therapy.

Carbonate apatite (CO_3_Ap) has a composition similar to bone, and it has attracted much attention as an alternative artificial bone graft material in recent years. Recently, the CO_3_Ap block was produced with a dissolution–precipitation reaction [17]. CO_3_Ap has morphological characteristics such as a large crystal size and surface area compared with HA and *β*-TCP [18]. An in vitro study showed that CO_3_Ap surfaces were more absorbed by rabbit osteoclast cells than HA [19]. In the culture of human bone marrow cells on CO_3_Ap, the expression level of osteoblast differentiation markers was significantly higher than HA and *β*-TCP [20]. In vivo studies reported that CO_3_Ap promoted new bone formation by inducing bone remodeling and angiogenesis [21,22]. The use of the CO_3_Ap granules was formally approved for implant and periodontal treatment in 2017 in Japan. The clinical effects of CO_3_Ap granules in dental implant therapy have been reported [23,24].

Combination therapy with signaling molecules and scaffold has been reported in the treatment of periodontitis [25,26,27]. In a randomized clinical trial and its follow-up studies, we have shown the clinical effectiveness of the use of rhFGF-2 and DBBM in the treatment of intrabony periodontal defects [9,10,14]. As for the combined use of FGF-2 and CO_3_Ap, it has been shown to enhance new bone formation in in vivo studies [28,29]. However, there remain uncertainties about the effectiveness of the combination of FGF-2 and CO_3_Ap on periodontal healing. Also, information is limited on its impact on cellular dynamics and osteoblast differentiation.

The aim of this study was to investigate in vivo and in vitro the effects of local application of FGF-2 in combination with CO_3_Ap on the healing of periodontal defects.

## 2. Materials and Methods

An overview of the conducted experiments is shown in Appendix A.

### 2.1. Animals

Forty-four Wistar rats (10-week-old male, weight of 250–300 g) were used (Sankyo Labo Service, Tokyo, Japan). The animals were housed in individual cages maintained under standard laboratory conditions and were given free access to standard laboratory rat chow and water. The experimental procedures were conformed to the Treatment of Experimental Animals at Tokyo Dental College (approval number 232205). This study adhered to the ARRIVE guidelines (https://arriveguidelines.org/, accessed on 5 January 2022).

### 2.2. FGF-2 and CO_3_Ap

Kaken Pharmaceutical (Tokyo, Japan) supplied the FGF-2. The bone substitute used was CO_3_Ap (Cytrans^®^ Granules, particle size 0.6–1.0 mm; GC, Tokyo, Japan). The FGF-2 used in the in vivo study was diluted with distilled water and hydroxypropyl cellulose (HPC; 3%) (FUJIFILM Wako Pure Chemical, Osaka, Japan) to generate a 0.3% solution of FGF-2. CO_3_Ap was loosely ground with a sterile mortar and pestle to adjust for filling the defect [30]. CO_3_Ap was premixed with FGF-2 and used after 10 min in the FGF-2 + CO_3_Ap group.

### 2.3. In Vivo Model and Surgical Protocol

The animals were allocated to the following groups: (1) Unfilled (*n* = 11), (2) FGF-2 (*n* = 11), (3) CO_3_Ap (*n* = 11), and (4) FGF-2 + CO_3_Ap (*n* = 11) (Appendix A). The animals underwent anesthesia via an intraperitoneal injection of a blend comprising midazolam (2 mg/kg), medetomidine (0.15 mg/kg), and butorphanol (2.5 mg/kg). Additionally, local infiltration anesthesia was administered. Following a crestal incision and flap elevation, periodontal defects (2.0 × 2.0 × 1.7 mm, width × length × depth) [31] were generated in the mesial aspect of the maxillary first molars (M1) (Appendix A). This was achieved utilizing a surgical template (Appendix A) [32]. The surgical procedure was implemented using a surgical scope. The root was meticulously stripped of its cementum and PDL. Subsequently, the sites were irrigated and dried. The defects in each group received FGF-2 solution (30 µL), CO_3_Ap (2 mg), or FGF-2 + CO_3_Ap or were left unfilled (Appendix A). The flaps were sutured with 6-0 resorbable sutures (Appendix A). Acetaminophen was then given.

### 2.4. Microcomputed Tomography

After 2 or 4 weeks, cardiovascular perfusion was conducted using paraformaldehyde (4%; FUJIFILM Wako Pure Chemical) following the administration of anesthesia to the animals. The maxillae were collected, and the healing of the defect area was assessed using a microcomputed tomography (micro-CT) system (R-mCT; Rigaku, Tokyo, Japan) (magnification, ×10; slice width, 16 µm). The region of interest (ROI) was delineated based on the following criteria: (1) longitudinally, from the alveolar bone crest at M1 mesial root to a depth of 1.7 mm, and (2) horizontally, standardized periodontal defects that included the proximal and buccal/palatal region of the defect. Data from the micro-CT were analyzed by image software (TRI/3D-BON; Ratoc System Engineering, Tokyo, Japan).

The bone mineral density (BMD) was measured as described previously [33]. New bone in the ROI was defined as the region with BMD ranging from 400 to 1000 mg/cm^3^, while CO_3_Ap and existing bone were categorized with BMD values exceeding 1000 mg/cm^3^ [34]. The newly formed bone was calculated by subtracting the regions for CO_3_Ap and bone from the total region with BMD > 400 mg/cm^3^.

The analysis focused on the bone volume fraction (BV/TV) in the ROI, excluding CO_3_Ap granules. In instances of therapy utilizing bone graft materials, assessing novel bone formation alone might lead to an underestimation of clinical success. Therefore, the ratio of the total radiopaque volume (comprising new bone and residual CO_3_Ap particles) to the total volume (RV/TV) was also examined [35,36].

### 2.5. Histological and Histomorphometric Analyses

The maxillae were bisected along the palatal median line for histological analysis. Following fixation in buffered 4% paraformaldehyde for 24 h, the samples were demineralized in 10% ethylenediaminetetraacetic acid disodium salt (EDTA-2Na, pH 7.0) (FUJIFILM Wako Pure Chemical) at 4 °C for a duration of 3 weeks and then embedded in paraffin. A microtome was used to cut 5 µm sections, and they underwent hematoxylin-eosin staining.

Histomorphometric analysis of new bone height was carried out using a microscope (UPM Axiophot 2; Carl Zeiss Japan, Tokyo, Japan) and software (Axio Vision 4.7; Carl Zeiss Japan). The relative level of the new bone was determined by the length from the apex to the crown cusp (Lac) of the first molar/the length of the bone gap (Lbg), the most coronal extent of newly formed bone to the level of intact alveolar bone (Appendix A) [37].

### 2.6. Immunohistochemistry

The preparation of the sections and the detection of osterix (Osx) and osteocalcin (OCN) were performed as described previously [36].

For quantitative analysis of the positive cells, an observation area was randomly assigned in each section. This analysis was conducted using image software (Image-Pro Plus 6.2, Media Cybernetics, Rockville, MD, USA) [38].

In each specimen, the quantification sites were categorized into three areas (the Bone side, the Middle area, and the Root side). The ratio of positive cells to total cells at each respective site was calculated [32,36,39].

The micro-CT, histomorphometric, and immunohistochemical data were assessed by one examiner, who was unaware of the grouping, and confirmed by a second examiner.

### 2.7. In Vitro Cell Culture

MC3T3-E1 cells (RIKEN BioResource Center, Tsukuba, Japan) were incubated in α-MEM (Gibco, Invitrogen, Carlsbad, CA, USA) containing 10% heat-inactivated fetal bovine serum and antimicrobials. The incubation was carried out at 37 °C in 5% CO_2_ in air.

CO_3_Ap was pre-incubated in the medium 5 days before to minimize alterations in the medium composition arising from ion leaching [40]. For the assessment of FGF-2-treated CO_3_Ap, following the complete removal of the medium, a solution of FGF-2 (5 µg/mL) was combined with the CO_3_Ap. The samples were incubated for 10 min at room temperature, and then the media containing cells were added.

### 2.8. Evaluation of Cell Behaviors

The morphology and spreading of the cells were evaluated using confocal laser scanning microscopy (CLSM) and scanning electron microscopy (SEM).

The preparation of samples for CLSM was carried out as described previously [36]. The prepared samples were imaged using a CLSM (LSM880, Carl Zeiss, Oberkochen, Germany) and ZEN 2 black software (Carl Zeiss). The samples for SEM were prepared using the method described previously [36]. An SU6600 SEM (Hitachi High-Tech Corporation, Tokyo, Japan) was used to characterize the cells.

### 2.9. Enzyme-Linked Immunosorbent Assay

The release of FGF-2 from the FGF-2-treated scaffold was assessed by enzyme-linked immunosorbent assay (ELISA). FGF-2 solution (100 µL) and CO_3_Ap (100 mg) were mixed for 10 min at room temperature. After rinsing, they were vortexed twice to eliminate non-absorbed FGF-2 [41]. They were incubated with PBS (500 µL) in a 24-well plate. The supernatant was collected, replaced with fresh PBS at each time point, and stored at −80 °C. The FGF-2 ELISA kit (R&D System, Minneapolis, MN, USA) was used for the measurements.

### 2.10. Cell Viability/Proliferation

MC3T3-E1 cells were seeded in 24-well plates (1 × 10^4^ cells/well) containing culture media and CO_3_Ap (100 mg), either with or without the presence of FGF-2. At 1, 3, and 5 days, the WST-8 (Cell Counting Kit-8; Dojindo Laboratories, Kumamoto, Japan) was performed following the manufacturer’s protocol.

### 2.11. Quantitative RT-PCR

The expression levels of the osteoblastic differentiation markers, *Runx2* and *Sp7*, were analyzed by quantitative RT-PCR (qRT-PCR) at 7 days. The total RNA of MC3T3-E1 cells cultured on the CO_3_Ap with/without FGF-2 was isolated with the RNeasy^®^ Mini Kit (Qiagen, Valencia, CA, USA) according to the manufacturer’s instructions. qRT-PCR analysis was performed using the 7500 Fast Real Time PCR system (Thermo Fisher Scientific, Waltham, MA, USA). The used primer sequences are shown in Table 1. The primers were designed using Genbank (https://www.ncbi.nlm.nih.gov/genbank/, accessed on 19 March 2022). Measurement of GAPDH served as an internal control. Relative gene expression levels were estimated using the 2^−∆∆Ct^ method.

### 2.12. Statistical Analysis

The sample size was estimated based on 90% power with a 0.05 two-sided significance level, given a 10.5% difference in bone volume between groups and a standard deviation of 7% [42]. For each group, a sample size of 10 (defect site) was needed at each time point. Considering the dropout (10%), the final sample size was set at *n* = 11.

Differences in data from the micro-CT and histomorphometric assessments were sought by analysis of variance (ANOVA) with the Tukey post hoc test. Comparisons for immunohistochemical data were analyzed by the Kruskal–Wallis test with Dunn’s post-test. In the WST-8 assay, intragroup comparisons were performed by the Kruskal–Wallis test with Dunn’s post hoc test. Comparisons for the WST-8 assay and qRT-PCR were performed by the Mann–Whitney U test. A software package (Prism ver 7.05; GraphPad Software, San Diego, CA, USA) was used. A *p*-value less than 0.05 was considered statistically significant.

## 3. Results

Four rats died during surgery. Primary wound closure was observed in all animals at 2 weeks postoperatively.

### 3.1. Micro-CT Analysis

No flap dehiscence was observed after suturing, and the healing process proceeded normally. There was no evidence of CO_3_Ap leakage.

Sagittal slice images from micro-CT obtained at the 2-week mark restricted new bone formation in the Unfilled group, and low BMD structures were observed around the CO_3_Ap (Figure 1a). Novel bone formation was observed adjacent to the root in the FGF-2 group. The presence of CO_3_Ap particles and the new bone formation were observed near the root within the CO_3_Ap and FGF-2 + CO_3_Ap groups.

Figure 1b illustrates the quantitative analysis of the newly formed bone (excluding CO_3_Ap within the ROI) at 2 weeks. The BV/TV values in the FGF-2 + CO_3_Ap group were significantly higher than those in the CO_3_Ap group, but there was no significant difference from the FGF-2 group. The values in the FGF-2, CO_3_Ap, and FGF-2 + CO_3_Ap groups were significantly higher than those in the Uunfilled group. The ratio of the total radiopaque volume (newly formed bone and residual CO_3_Ap) to the total volume (RV/TV) was assessed (Figure 1c). The RV/TV values in the CO_3_Ap and FGF-2 + CO_3_Ap groups were higher than those in the Unfilled or FGF-2 groups.

At 4 weeks, new bone formation remained restricted in the Unfilled group, while there was an apparent enhancement in the other experimental groups (Figure 1d). The BV/TV values in the FGF-2 + CO_3_Ap group were significantly higher than those in the CO_3_Ap group, but there was no significant difference from the FGF-2 group. The values in the FGF-2, CO_3_Ap, and FGF-2 + CO_3_Ap groups were significantly higher than those in the Unfilled group (Figure 1e). The RV/TV values in the CO_3_Ap and FGF-2 + CO_3_Ap groups were significantly higher than those in the Unfilled or FGF-2 groups (Figure 1f).

### 3.2. Histological Analysis

At 2 and 4 weeks postoperatively, in all groups, the previous defect area was filled with newly formed connective tissue and vascular-like structures, while different levels of new bone formation was observed near the Root side(Figure 2). At 2 weeks, CO_3_Ap granules were observed in the CO_3_Ap and FGF-2 + CO_3_Ap groups (Figure 2c,d,i,j). Fibrous connective tissue surrounded the CO_3_Ap granules, with the presence of multinucleated giant cells (Figure 2e,f). The FGF-2 applied groups showed a tendency for more pronounced vascular-like structures compared with the non-applied groups at 2 weeks (Figure 2a–d).

In the FGF-2, CO_3_Ap, and FGF-2 + CO_3_Ap groups, more progressed new bone formation was observed along the root than the Unfilled group at 2 and 4 weeks (Figure 2a–d,g–j). At 4 weeks, in the FGF-2 + CO_3_Ap group, the level of newly formed bone was the greatest among groups, and the CO_3_Ap particles positioned near the root of the previous defect were incorporated in the newly formed bone (Figure 2l). There were no signs of ankylosis.

In the histomorphometric analysis of relative new bone height, the FGF-2, CO_3_Ap, and FGF-2 + CO_3_Ap groups exhibited significantly greater values than the Unfilled group at 2 and 4 weeks (Appendix A). At 2 weeks, the FGF-2 and FGF-2 + CO_3_Ap groups showed significantly higher levels than the CO_3_Ap group (Appendix A). At 4 weeks, the FGF-2 + CO_3_Ap group demonstrated significantly higher levels compared with the FGF-2 and CO_3_Ap groups (Appendix A).

### 3.3. Immunohistochemical Analyses

At 2 weeks, Osx-positive cells were frequently observed on the surface of newly formed bonein the Root side (Figure 3a,d), as well as on the existing bone in the Bone side (Figure 3b,e) and around the CO_3_Ap particles.

Regarding the results from the quantitative analysis at 2 weeks postoperatively, in the Root side, the ratio of Osx-positive cells in the FGF-2 + CO_3_Ap group was found to be higher than that in both the Unfilled and CO_3_Ap groups (Table 2). On the Bone side, the ratio in both the CO_3_Ap and FGF-2 + CO_3_Ap groups exceeded that in the Unfilled group.

Within the Middle area (Figure 3c), the FGF-2 + CO_3_Ap group exhibited a higher ratio compared with the Unfilled group. In intragroup comparisons, within the FGF-2 group, the ratio of Osx-positive cells on the Root side was significantly higher compared with the Bone side and the Middle area. In the CO_3_Ap and FGF-2 + CO_3_Ap groups, within the Root side, the ratio was significantly higher compared with the Middle area.

At 4 weeks, there appeared to be a higher presence of Osx-positive cells on the existing bone surface of the Bone side (Figure 3e) compared with that at 2 weeks (Figure 3b). The ratio of Osx-positive cells in the FGF-2 + CO_3_Ap group was significantly higher than the Unfilled group in the Root side, Bone side, and Middle area (Figure 3f, Table 2). In intragroup comparisons, within the unfilled group, the ratio of Osx-positive cells on the Bone side was significantly higher compared with the Middle area (Table 2). Additionally, in the FGF-2 and CO_3_Ap groups, the ratio of Osx-positive cells on both the Root side and the Bone side was significantly higher than the Middle area.

Furthermore, in the Unfilled, FGF-2, CO_3_Ap, and FGF-2 + CO_3_Ap groups, the ratio of Osx-positive cells exhibited an increasing trend compared with the 2 weeks measurement in all area. The Bone side demonstrated the most pronounced increase in positive cell ratio.

OCN-positive cells were frequently observed within the newly formed bone on the Root side (Figure 4a,d), within the existing bone on the Bone side (Figure 4b,e), and on the surface of the CO_3_Ap particles. The quantitative analysis revealed that at 2 weeks postoperatively, the ratio of OCN-positive cells in the Root side was significantly higher in the FGF-2 + CO_3_Ap group compared with the Unfilled group (Table 2). Furthermore, in the Bone side (Figure 4b) and in the Middle area (Figure 4c), the FGF-2 group exhibited a significantly higher the ratio of OCN-positive cells than the Unfilled group (Table 2). The FGF-2 + CO_3_Ap group showed a significant increase compared with the Unfilled and CO_3_Ap groups. In intragroup comparisons, in the CO_3_Ap group, the ratio of OCN-positive cells in the Root side was significantly higher than the Bone side and Middle area (Table 2).

At 4 weeks, in the Root side (Figure 4d), the ratio of OCN-positive cells in the FGF-2 + CO_3_Ap group was significantly higher than the Unfilled group (Table 2). In the Bone side (Figure 4e), the ratio of OCN-positive cells in the FGF-2 group was significantly higher compared with the Unfilled and CO_3_Ap groups (Table 2). In the Middle area (Figure 4f), the FGF-2 group showed a significant increase compared with the Unfilled group, while the FGF-2 + CO_3_Ap group showed a significant increase compared with the Unfilled and CO_3_Ap groups. In intragroup comparisons, the ratio of OCN-positive cells in the Bone side was significantly higher than the Middle area in the Unfilled and FGF-2 groups. Additionally, for the CO_3_Ap and FGF-2 + CO_3_Ap groups, the ratio of OCN-positive cells in the Root side was significantly higher compared with the Middle area. The ratio of OCN-positive cells exhibited an increasing trend at 4 weeks compared with 2 weeks postoperatively. In the CO_3_Ap and FGF-2 + CO_3_Ap groups, there was a tendency for a higher OCN-positive ratio in the Root side (Figure 4d), while in the Unfilled and FGF-2 groups, the Bone side (Figure 4e) showed a tendency for a higher OCN-positive ratio.

### 3.4. In Vitro Cell Morphology and Spreading

SEM images showed MC3T3-E1 cells adhering to the CO_3_Ap and FGF-2-treated CO_3_Ap surfaces (Figure 5a–d). In the FGF-2 + CO_3_Ap group, a greater level of cell attachment was observed compared with the CO_3_Ap group. FGF-2 induced a greater level of cell attachment and more elongated cells compared with the non-applied group (Figure 5c,d).

Similarly, analysis of CLSM images revealed that a greater number of cells adhered to and spread on the scaffold in the FGF-2 + CO_3_Ap group, when compared with the CO_3_Ap group (Figure 5e,f).

### 3.5. FGF-2 Release

When the FGF-2 release kinetics from the FGF-2-treated CO_3_Ap was assessed using ELISA, FGF-2 continued to be released up to 120 h, reaching approximately 682 pg/mL in total release (Appendix A), which was approximately 92% of the total amount released.

### 3.6. Cell Viability/Proliferation

The FGF-2 + CO_3_Ap group exhibited significantly higher viability/proliferation compared with the CO_3_Ap group at 1, 3, and 5 days of incubation (Figure 6). The CO_3_Ap and FGF-2 + CO_3_Ap groups showed a significantly higher viability/proliferation at 5 days compared with one day.

### 3.7. Expression of Runx2 and Sp7

The CO_3_Ap group demonstrated significantly higher gene expression levels of *Runx2* and *Sp7* than the FGF-2 + CO_3_Ap group at 7 days (Figure 7).

## 4. Discussion

In micro-CT analysis, FGF-2, CO_3_Ap, and FGF-2 + CO_3_Ap groups exhibited significantly higher BV/TV values compared with the Unfilled group. The FGF-2 + CO_3_Ap group showed significantly higher BV/TV values compared with the CO_3_Ap group. Histomorphometric observations revealed that the FGF-2 + CO_3_Ap group exhibited the highest levels of new bone. In previous studies, when CO_3_Ap was applied to periodontal defects in dogs, the amount of newly formed bone was significantly higher in the applied group compared with the non-applied group [22,29]. These results further support the osteoconductive property of CO_3_Ap and suggest that the addition of FGF-2 enhances bone formation when combined with CO_3_Ap.

When employing combination periodontal therapy with bone substitutes, it is important to note that micro-CT analysis of bone formation might underestimate its effect. This is attributed to the fact that the presence of residual material can restrict the volume data of new bone. In clinical studies, both newly formed bone and the remnants of bone substitutes are frequently employed as a metric for assessing outcomes [9,10,14]. Thus, we further assessed the total radiopaque volume (new bone + CO_3_Ap) per total volume (RV/TV). The RV/TV values were found to be significantly greater in the CO_3_Ap and FGF-2 + CO_3_Ap groups compared with the Unfilled and FGF-2 groups. This result was similar to the findings from the previous study using FGF-2 and DBBM [36]. In a preclinical study, the application of CO_3_Ap to periodontal defects resulted in an increase in the amount of newly formed bone [43]. These results suggest that CO_3_Ap functions as a scaffold in periodontal healing, and the elevated bone levels in the CO_3_Ap and FGF-2 + CO_3_Ap groups were attributed not only to the presence of CO_3_Ap but also to the initiation of novel bone formation.

Each treated site was divided into three areas to investigate the details of the healing processes by immunohistochemical analyses. At 2 weeks, Osx and OCN-positive cells were frequently seen in the Root side in the CO_3_Ap and FGF-2 + CO_3_Ap groups. On the other hand, in the FGF-2 group, Osx-positive cells were most prominently observed on the Root side, while OCN-positive cells were more abundant on the Bone side. Osx is recognized as an early marker for osteoblast differentiation, while OCN is considered a marker for later stages of differentiation [44,45]. At 4 weeks postoperatively, Osx and OCN-positive cells increased in all areas compared with at 2 weeks. Interestingly, Osx-positive cells were highly prevalent on the Bone side, while OCN-positive cells exhibited the highest level on the Root side in the CO_3_Ap and FGF-2 + CO_3_Ap groups. In the FGF-2 group, Osx and OCN-positive cells were more prominent on the Bone side compared with other areas, at 2 and 4 weeks. In a canine model, FGF-2 has been demonstrated to enhance cell proliferation and induce osteoblast differentiation after one week [46]. Furthermore, it has been reported that the cultivation of human bone marrow stem cells on CO_3_Ap results in a higher expression of osteoblast differentiation markers [20]. Based on these findings, it is suggested that CO_3_Ap has potential as a scaffold as well as carrier for FGF-2 during the observation period. Furthermore, adding CO_3_Ap to FGF-2 may promote cell proliferation in the periodontal defects, particularly in the Root side, which consequently contributed to osteoblastic differentiation.

In vitro, the addition of FGF-2 led to the observation of a greater number of cells, and these MC3T3-E1 cells appeared more elongated. The cells cultured on the CO_3_Ap with the addition of FGF-2 exhibited a significantly higher cell viability/proliferation compared with those cultured on CO_3_Ap. FGF-2 has been reported to enhance cell proliferation by increasing the expression of CD44, which is involved in cell adhesion and proliferation [47]. These findings suggest that FGF-2 promotes initial cell attachment and spreading on CO_3_Ap, contributing to cell proliferation. In our previous study, PDLCs on the FGF-2-treated DBBM exhibited increased viability/proliferation and showed longer lamellipodia-like cell protrusions than non-treated DBBM [36]. CO_3_Ap releases calcium ions, which promote the differentiation and proliferation of osteoblasts [48]. Furthermore, the addition of calcium ions to the Ti disk has been reported to increase cell adhesion within 24 h post-addition [49]. The release of calcium ions from CO_3_Ap may have facilitated the cell attachment. Taken together, it is possible that CO_3_Ap is a suitable scaffold to be used with FGF-2. Further investigations are needed to identify the most appropriate scaffold or carrier to be used with FGF-2.

The evaluation of FGF-2 release kinetics from FGF-2-treated CO_3_Ap revealed sustained release, over a period of 120 h. This finding was similar to findings from previous studies that investigated the addition of FGF-2 to bone graft materials [28,36]. The adsorption and sustained release of growth factors onto the surface of biomaterials have a profound influence on cell proliferation and osteoblast differentiation [50]. In vivo, FGF-2 exhibits a short half-life [51]. The increased cell proliferation and osteoblastic differentiation observed in the FGF-2-treated CO_3_Ap are presumed to be attributable to the release of FGF-2 adsorbed onto CO_3_Ap.

The addition of FGF-2 resulted in a decreased expression levels of *Runx2* and *Sp7* in MC3T3-E1 cells cultured on CO_3_Ap at 7 days. During the early stage of periodontal tissue healing, FGF-2 promotes cell proliferation in PDLCs while inhibiting their differentiation, and in the late phase, it facilitates their differentiation into osteoblast [3]. The in vitro and in vivo experiments conducted in this study yielded similar findings to those from the previous study. Taken together, it is suggested that FGF-2 promoted initial attachment to and proliferation on CO_3_Ap and regulated osteoblastic differentiation (suppression at the early phase and promotion at the later phase), which contributed to novel bone formation in the previous defect.

This study has several limitations. The grouping of in vivo and in vitro cell experiments was different. This is because the cell behaviors in a two-dimensional (2D) culture are different from those in a three-dimensional (3D) culture. Therefore, only the groups with the scaffold are used for in vitro experiments. Additional in vitro experiments are needed to prove the ability of the combination therapy to promote bone formation. To determine whether the combined use of FGF-2 and CO_3_Ap promotes overall periodontal regeneration, additional experiments with an extended observation period are necessary. Furthermore, a thorough understanding of the precise mechanism of the combined effects requires more detailed investigations at the molecular and genetic level.

However, the results from this study provide salient insight into the effectiveness of the combination regenerative therapy.

## 5. Conclusions

The combined application of FGF-2 and CO_3_Ap yielded enhanced healing in the periodontal defect. The addition of FGF-2 to CO_3_Ap promoted initial cell attachment to and proliferation on CO_3_Ap and regulated osteoblastic differentiation, thereby contributing to new bone formation from near the root.


**Clinical significance**


⮚Information is still limited regarding the performance of combination regenerative therapy using FGF-2 and CO_3_Ap.⮚Combined use of FGF-2 and CO_3_Ap yielded enhanced healing in the surgically created periodontal defects.⮚Biofunctionalization of CO_3_Ap with FGF-2 may enhance periodontal healing via promotion of cell proliferation, angiogenesis, and regulation of osteogenic differentiation.

## Figures and Tables

**Figure 1 biomedicines-12-01664-f001:**
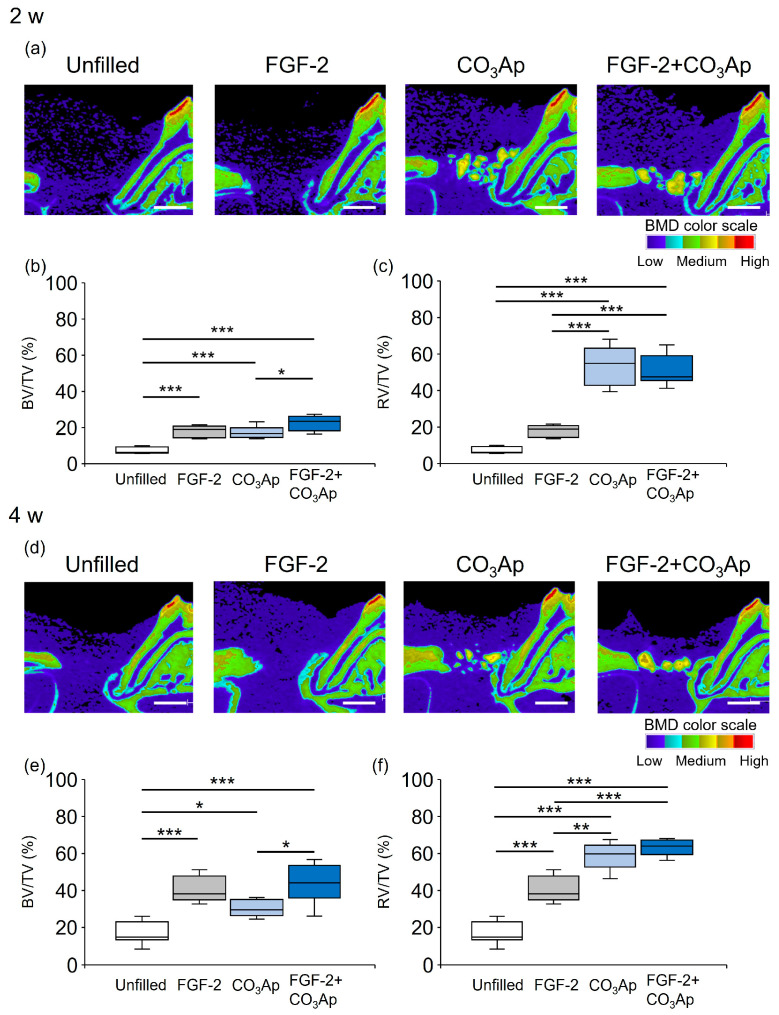
Micro-CT images and quantification analysis. The color scale indicates bone mineral density (BMD): light blue and purple, low; yellow and green, medium; and red and orange, high. (**a**,**d**) Sagittal images from micro-CT (bar = 1000 µm). (**b**,**c**,**e**,**f**) Quantitative data. Bone volume (BV)/total volume (TV) within the ROI (**b**,**e**) and radiopaque volume of newly formed bone and CO_3_Ap particles (RV)/total volume (TV) (**c**,**f**) were compared between groups. Data are presented as box-and-whiskers plots with maximum, median, minimum, and 75th and 25th percentiles (*n* = 10). * *p* < 0.05, ** *p* < 0.01, *** *p* < 0.001 by ANOVA with Tukey post hoc test.

**Figure 2 biomedicines-12-01664-f002:**
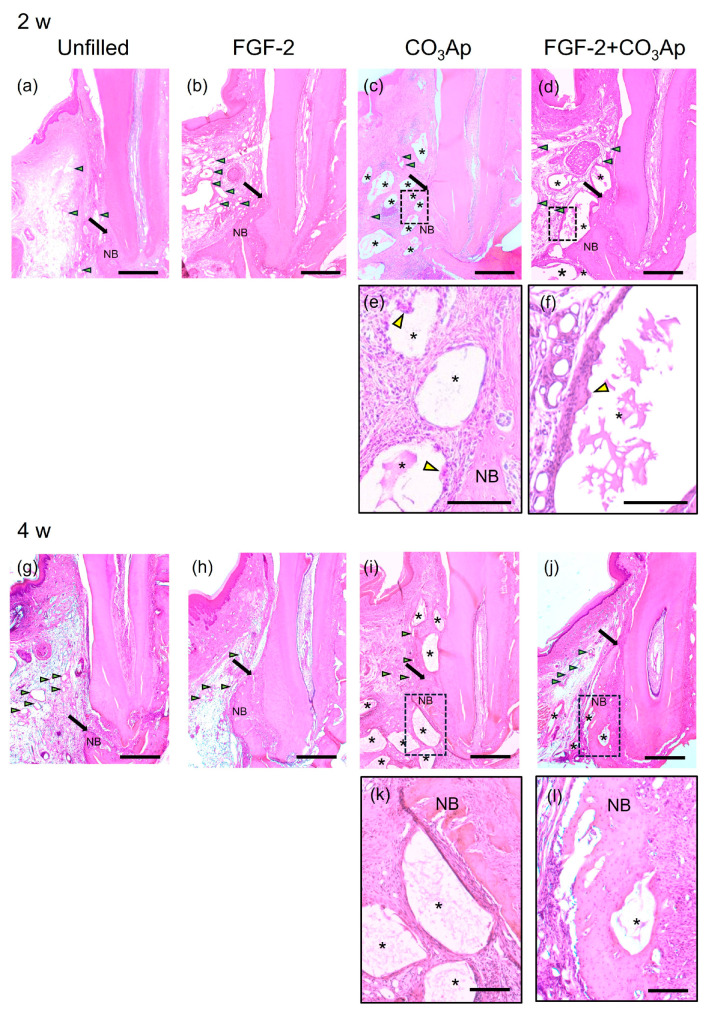
Histopathological analysis. (**a**–**d**) Images at 2 weeks (original magnification ×25; bar = 500 µm; green arrowheads indicate the blood vessels; black arrows indicate the most coronal position of newly formed bone; yellow arrowheads indicate multinucleated giant cell; NB, newly formed bone; asterisk indicates CO_3_Ap particle). (**a**) The Unfilled group shows minimal new bone formation. (**b**–**d**) In the FGF-2, CO_3_Ap, and FGF-2 + CO_3_Ap groups, novel bone can be observed in the Root side of the intrabony defect. (**e**,**f**) Higher-magnification images of the framed area in the corresponding group (original magnification ×100; scale bar = 200 µm). Fibrous connective tissue surrounded the CO_3_Ap granules, with the presence of a multinucleated giant cell observed. (**g**–**j**) Images at 4 weeks (original magnification ×25; bar = 500 µm). (**e**) The Unfilled group exhibits limited new bone formation. (**h**–**j**) Newly formed bone in the FGF-2, CO_3_Ap, and FGF-2 + CO_3_Ap groups appears to be greater compared with the Unfilled group. (**k**,**l**) At 4 weeks, an enlarged image of the framed area was captured (original magnification ×100; scale bar = 200 µm).

**Figure 3 biomedicines-12-01664-f003:**
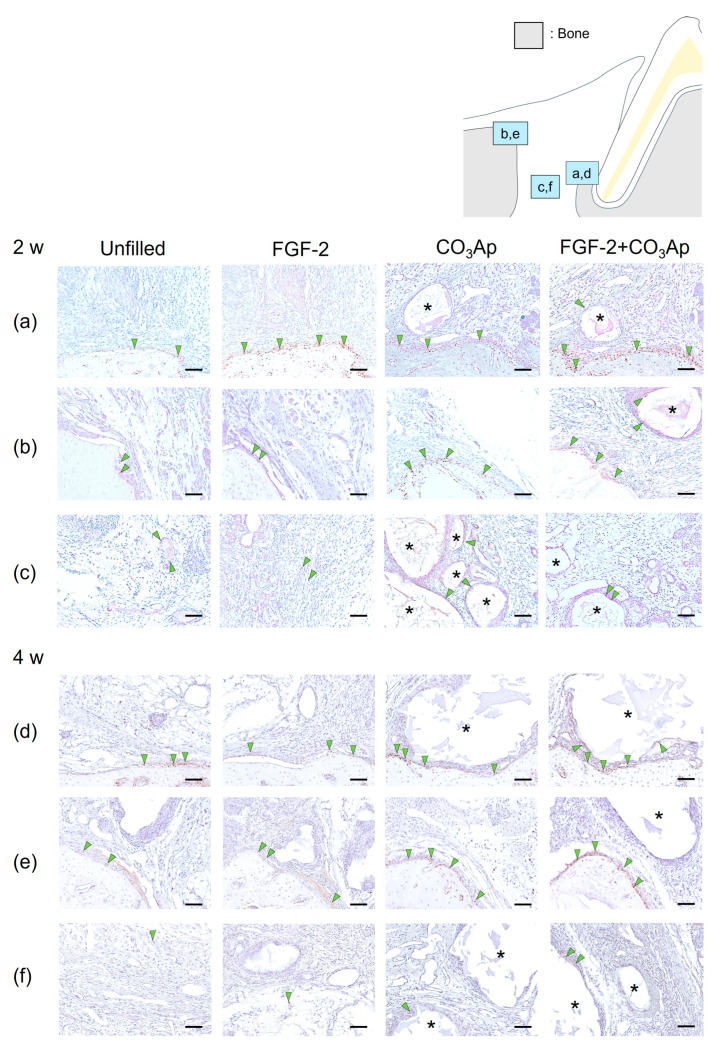
Immunohistochemistry for Osx. Positive cells are assessed in the Root side (**a**,**d**), Bone side (**b**,**e**), and Middle area (**c**,**f**). A brown coloration shows an Osx-positive reaction (arrowheads indicate the general location of the positive cells). At 2 weeks (**a**–**c**), the Osx-positive cells were observed in new bone and existing bone and around CO_3_Ap. At 4 weeks (**d**–**f**), the number of Osx-positive cells in the Root side and Bone side appears to be greater in the FGF-2 + CO_3_Ap group compared with the Unfilled group. (Osx and counterstaining with Mayer’s hematoxylin stain, original magnification ×200; bar = 50 µm; asterisk shows CO_3_Ap particles.)

**Figure 4 biomedicines-12-01664-f004:**
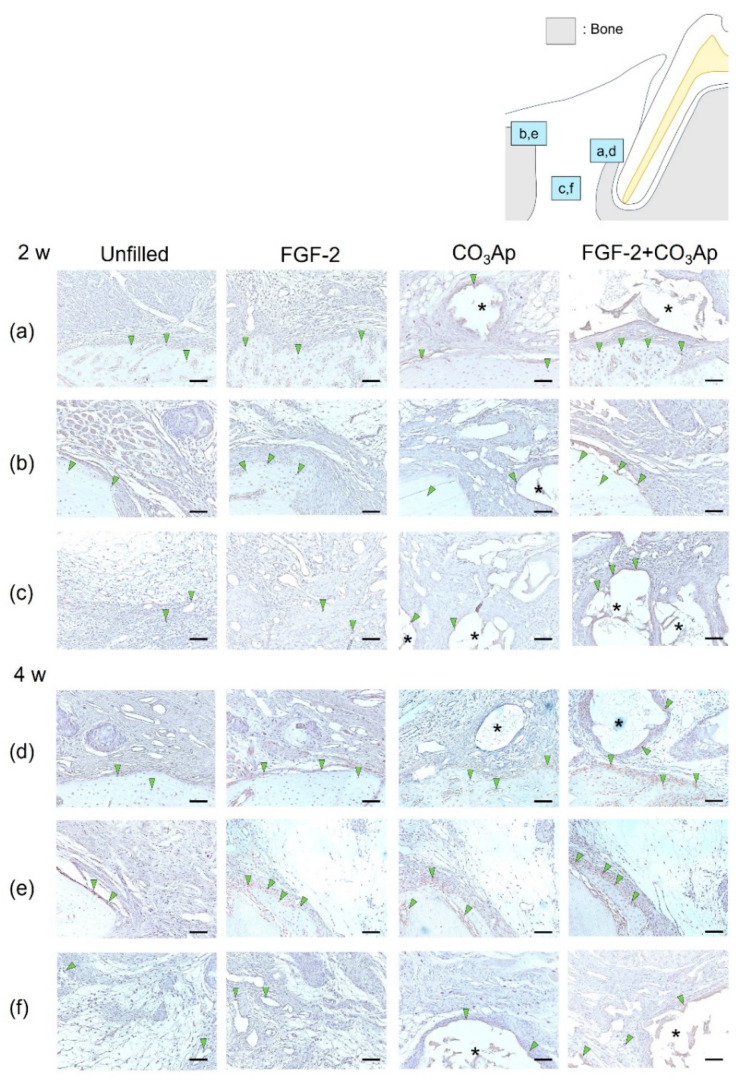
Immunohistochemical staining for OCN. Prevalence of OCN-positive cells is assessed in the Root side (**a**,**d**), Bone side (**b**,**e**), and Middle area (**c**,**f**). A brown coloration shows an OCN-positive reaction (arrowheads indicate the general location of the positive cells). At 2 weeks (**a**–**c**), the OCN-positive cells were observed in new bone, existing bone, and CO_3_Ap. At 4 weeks (**d**–**f**), the number of OCN-positive cells in the Root side and Middle area appears to be greater in the FGF-2 + CO_3_Ap groups compared with the Unfilled group. (OCN and counterstaining with Mayer’s hematoxylin stain, original magnification ×200; bar = 50 µm; asterisk indicates CO_3_Ap particles.)

**Figure 5 biomedicines-12-01664-f005:**
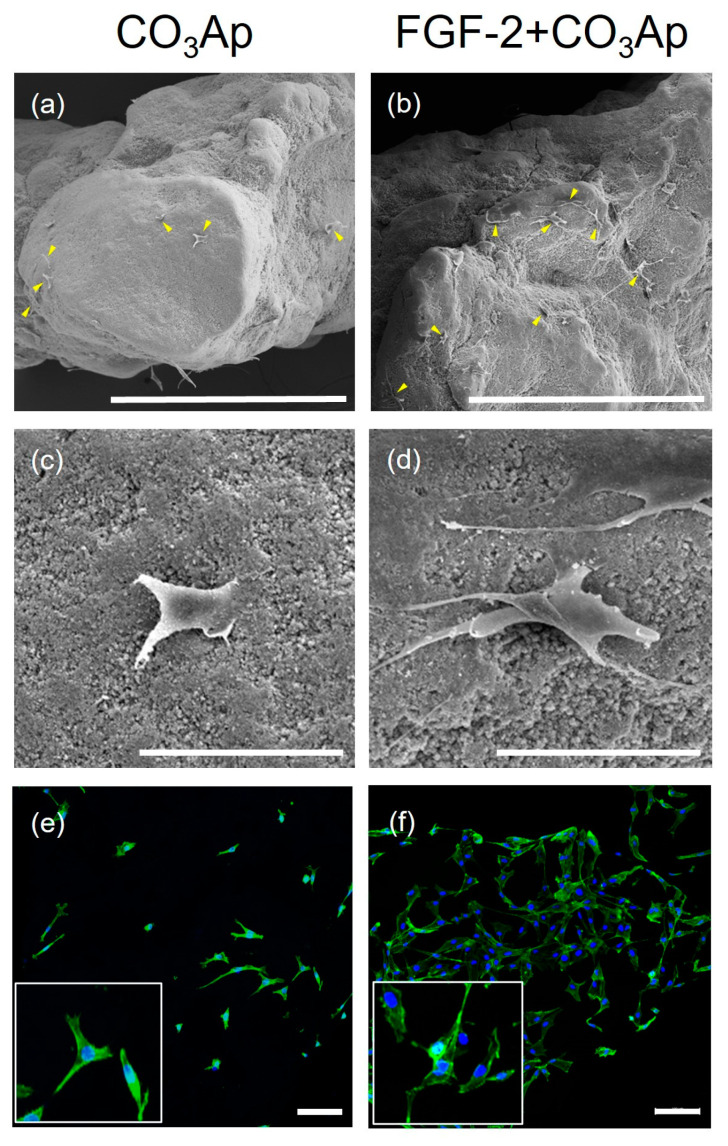
MC3T3-E1 cells cultured on the CO_3_Ap with/without FGF-2. SEM images show that MC3T3-E1 attached to the CO_3_Ap (**a**,**c**) and FGF-2 + CO_3_Ap (**b**,**d**) at 24 h. (**a**–**d**) A greater number of cells appear to be attached to the FGF-2-treated CO_3_Ap compared with CO_3_Ap. In the FGF-2 + CO_3_Ap group, cell protrusions are more evident compared with the CO_3_Ap group. (**a**,**b**) Original magnification ×130; bar = 400 μm. (**c**,**d**) Original magnification ×1000; bar = 50 μm. Yellow arrowheads indicate MC3T3-E1. CLSM images reveal cells stained for actin (green) and the nucleus (blue) at 24 h (**e**,**f**). Higher-magnification images are shown in the insets. Compared with the CO_3_Ap group (**e**), a greater number of attached cells are observed in the FGF-2 + CO_3_Ap group (**f**). (Original magnification ×100; bar = 100 μm.)

**Figure 6 biomedicines-12-01664-f006:**
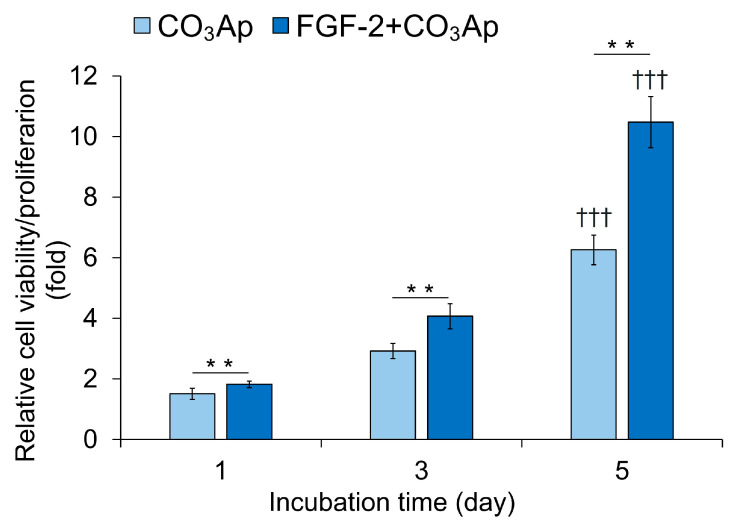
Viability/proliferation of MC3T3-E1 cells. Cells were seeded onto the CO_3_Ap with/without FGF-2 in the culture media and allowed to grow for up to 5 days. The WST-8 assay was employed to assess cell viability and proliferation at the indicated time points. The reference absorbance at 450 nm was subtracted from the absorbance of each sample, and the resulting values were expressed relative to those at 0 h. The FGF-2 + CO_3_Ap group exhibited significantly higher viability/proliferation compared with the CO_3_Ap group at 1, 3, and 5 days. Data are shown as mean ± SD (*n* = 6). ** *p* < 0.01, by Mann–Whitney U test. ††† *p* < 0.001 significant difference from one day values by Kruskal–Wallis test with Dunn’s post hoc test.

**Figure 7 biomedicines-12-01664-f007:**
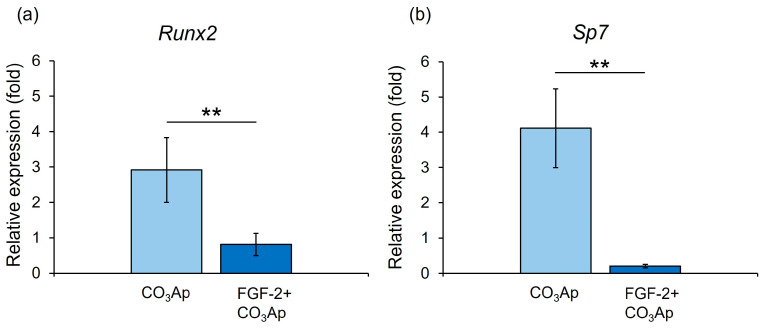
qRT-PCR assessment of the expression levels of *Runx2* and *Sp7*. Relative *Runx2* (**a**) and *Sp7* (**b**) expression levels in MC3T3-E1 cells on CO_3_Ap with/without FGF-2 at 7 days. The CO_3_Ap group demonstrated significantly higher expression levels of *Runx2* and *Sp7* compared with the FGF-2 + CO_3_Ap group at 7 days. The resulting values are expressed relative to the control group (control group; culture medium only). Data are presented as mean ± SD (*n* = 6). ** *p* < 0.01, by Mann–Whitney U test.

**Table 1 biomedicines-12-01664-t001:** Primer sequences used for the qRT-PCR in this study.

*Runx2*	F: 5′-GCCCAGGCGTATTTCAGA-3′ R: 5′-TGCCTGGCTCTTCTTACTGAG-3′
*Sp7*	F: 5′-CTCCTGCAGGCAGTCCTC-3′ R: 5′-GGGAAGGGTGGGTAGTCATT-3′
*GAPDH*	F: 5′-ATCACCATCTTCCAGGAG-3′ R: 5′-ATGGACTGTGGTCATGAG-3′

**Table 2 biomedicines-12-01664-t002:** Quantitative analysis of the Osx and OCN-positive cells.

	Area/Group		Unfilled	FGF-2	CO_3_Ap	FGF-2 + CO_3_Ap
Osx	2 w	Root side	7.8 ± 5.8	16.2 ± 6.6 *^,†^	18.4 ± 4.4 *	27.0 ± 4.0 ^a,b,^*
		Bone side	3.2 ± 1.5	6.6 ± 2.4	12.8 ± 7.0 ^a^	22.0 ± 10.5 ^a^
		Middle area	3.8 ± 1.3	6.2 ± 2.5	8.6 ± 2.6	13.4 ± 3.6 ^a^
	4 w	Root side	8.3 ± 2.5	18.6 ± 3.9 *	18.5 ± 2.5 *	27.8 ± 5.1 ^a,^*
		Bone side	13.3 ± 3.4 *	19.6 ± 1.5 *	18.5 ± 6.5 *	22.4 ± 6.5 ^a^
		Middle area	5.9 ± 1.7	11.6 ± 2.7	9.7 ± 2.2	13.8 ± 1.0 ^a^
OCN	2 w	Root side	6.1 ± 2.1	10.5 ± 2.7	10.3 ± 1.7 *^,†^	18.1 ± 4.0 ^a^
		Bone side	5.6 ± 1.9	12.6 ± 2.5 ^a^	6.9 ± 1.9	14.7 ± 3.5 ^a,b^
		Middle area	6.7 ± 1.6	12.0 ± 1.9 ^a^	6.6 ± 2.0	13.5 ± 2.7 ^a,b^
	4 w	Root side	10.7 ± 1.7	16.7 ± 3.1	15.7 ± 2.2 *	23.4 ± 3.2 ^a,^*
		Bone side	13.9 ± 3.9 *	20.3 ± 1.1 ^a,b,^*	12.6 ± 2.0	16.9 ± 1.5
		Middle area	8.5 ± 0.5	11.9 ± 1.1 ^a^	9.4 ± 1.9	14.1 ± 1.1 ^a,b^

Positive cells/total cells (%) in the target area. ^a^ Significantly different from the Unfilled group. ^b^ Significantly different from the CO_3_Ap group. * Significantly different from the Middle area. ^†^ Significantly different from the Bone side. Data are shown as mean ± SD (*n* = 6) by the Kruskal–Wallis test with Dunn’s post hoc test (*p* < 0.05–0.001).

## Data Availability

The data that support the findings of this study are available from the corresponding author upon reasonable request.

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
