# Peer review of "Combined Effects of Fibroblast Growth Factor-2 and Carbonate Apatite Granules on Periodontal Healing: An In Vivo and In Vitro Study"

_biomedicines, 2024, doi:10.3390/biomedicines12081664_

Round 1
Reviewer 1 Report
Comments and Suggestions for Authors
The reviewer appreciates the effort of the author in completing a nice study. The research is well-designed, conducted, and executed. The manuscript is well-written without any major issues in it. However, after careful review, the reviewer has only a few recommendations
Add a methodological illustration for easy understanding of the research design
Add a few clinical significances in bullet points at the end of the conclusions
Author Response
Reviewer 1
The reviewer appreciates the effort of the author in completing a nice study. The research is well-designed, conducted, and executed. The manuscript is well-written without any major issues in it. However, after careful review, the reviewer has only a few recommendations.
(Response)
We thank the reviewer for such an encouraging comment.
We revised the manuscript according to the reviewer’s advice.
Add a methodological illustration for easy understanding of the research design
(Response)
We have provided a methodological illustration in the Supplemental Figure. However, in light of the reviewer’s advice, we added a methodological illustration showing the overview of the in vivo and in vitro experiments. We provide this as a supplemental figure.
Add a few clinical significances in bullet points at the end of the conclusions
(Response)
Thank you for this advice. We added the clinical significances in bullet points at the end of the conclusions.
Reviewer 2 Report
Comments and Suggestions for Authors
In the article entitled " Combined Effects of Fibroblast Growth Factor-2 and Carbonate 2 Apatite Granules on Periodontal Healing: An In Vivo 3 and In Vitro Study ", the author introduced the effects of FGF-2 and CO3Ap complexes on periodontal healing in vivo. It can significantly increase the adhesion and proliferation of cells, regulate osteogenic differentiation and promote the formation of new bone through the repair of suitable materials. This material provides a new therapeutic strategy for guiding periodontal bone regeneration. However, a few minor issues need to be revised before publication:
1. For biocompatibility tests, the author only conducted cell proliferation tests, and suggested that related tests such as AM/PI staining tests should be added to further demonstrate the biocompatibility of FGF-2 and CO3Ap complexes.
2. The author's grouping of animal experiments and cell experiments is different, please supplement the grouping and related experiments.
3. The authors used tables to analyze the quantitative analysis of OSX and ocn positive cells, but the differences could not be seen directly. Please change them into pictures to visually observe the differences in the expression of OSX and OCN.
4. In vitro cell experiments, the author did not use relevant experiments to prove the ability of composite materials to promote bone formation in vitro tests, please supplement relevant experiments.
5. To more clearly elaborate on the problem, the introduction should be retouched and modified. Most recent studies such as Biomaterials Translational, 10.12336/biomatertransl.2024.02.002. Biomaterials Translational, 10.12336/biomatertransl.2024.02.003. Biomaterials Translational, 10.12336/biomatertransl.2024.02.003. Cell Rep Med, 10.1016/j.xcrm.2024.101574.
Author Response
Reviewer 2
In the article entitled " Combined Effects of Fibroblast Growth Factor-2 and Carbonate 2 Apatite Granules on Periodontal Healing: An In Vivo 3 and In Vitro Study ", the author introduced the effects of FGF-2 and CO3Ap complexes on periodontal healing in vivo. It can significantly increase the adhesion and proliferation of cells, regulate osteogenic differentiation and promote the formation of new bone through the repair of suitable materials. This material provides a new therapeutic strategy for guiding periodontal bone regeneration. However, a few minor issues need to be revised before publication:
(Response)
We appreciate the reviewer for the positive comments.
- For biocompatibility tests, the author only conducted cell proliferation tests, and suggested that related tests such as AM/PI staining tests should be added to further demonstrate the biocompatibility of FGF-2 and CO3Ap complexes.
(Response)
We thank the reviewer for the important advice.
WST-8 we employed is a test not only for cell proliferation but also for cell viability. However, we would like to include AM/PI staining test in our future studies. Biocompatibility itself of FGF-2 or CO3Ap has been shown in the previous studies (Murakami S et al. https://doi.org/10.1111/j.1600-0757.2010.00365.x, Yotsova R and Peev S. https://doi.org/10.3390/pharmaceutics16020291, Nifant’ev IE et al. https://doi.org/10.1021/acsabm.3c00753)
In a small-scale clinical study, the combined use of FGF-2 and CO3Ap has been shown to be clinically safe (Kitamura M et al. https://doi.org/10.1016/j.reth.2022.06.001).
- The author's grouping of animal experiments and cell experiments is different, please supplement the grouping and related experiments.
(Response)
We appreciate this comment. Indeed, the groupings of in vivo experiments and in vitro experiments are different. One reason for this is the difference in cell behaviors between 2D and 3D culture conditions. Because we cannot directly compare the cell groups cultured without the scaffold (2D culture) with those with the scaffold (3D culture), it was not possible to apply the same groupings in the in vitro and in vivo experiments.
In light of the reviewer’s concern, we added this as a limitation of the study, in the revised manuscript.
- The authors used tables to analyze the quantitative analysis of OSX and ocn positive cells, but the differences could not be seen directly. Please change them into pictures to visually observe the differences in the expression of OSX and OCN.
(Response)
We provided the results from the immunohistochemical analysis in the figures as well as the table. However, the figure for OCN was provided as a supplemental figure. In light of the reviewer’s comments, we moved the figure to the main text so that the differences could be directly seen.
- In vitro cell experiments, the author did not use relevant experiments to prove the ability of composite materials to promote bone formation in vitro tests, please supplement relevant experiments.
(Response)
Although this study was designed as in vivo and in vitro study, a particular focus was placed on the in vivo experiments. We acknowledge the need for additional in vitro experiments, such as Alp and alizarin red staining, qRT-PCR for later stage osteogenic differentiation markers, to prove the ability of the combination therapy to promote bone formation. We would like to include those in our future study. We added this as another limitation of the study in the revised manuscript.
- To more clearly elaborate on the problem, the introduction should be retouched and modified. Most recent studies such as Biomaterials Translational, 10.12336/biomatertransl.2024.02.003. Cell Rep Med, 10.1016/j.xcrm.2024.101574.
(Response)
We thank the reviewer for the important advice. We referred to the suggested papers and modified the Introduction section of the manuscript to more clearly elaborate on the problem.
Round 2
Reviewer 2 Report
Comments and Suggestions for Authors
The manuscript has been well revised and could be accepted.